# Antioxidants N-Acetylcysteine and Vitamin C Improve T Cell Commitment to Memory and Long-Term Maintenance of Immunological Memory in Old Mice

**DOI:** 10.3390/antiox9111152

**Published:** 2020-11-19

**Authors:** Andreas Meryk, Marco Grasse, Luigi Balasco, Werner Kapferer, Beatrix Grubeck-Loebenstein, Luca Pangrazzi

**Affiliations:** 1Immunology group, Institute for Biomedical Aging Research, University of Innsbruck, 6020 Innsbruck, Austria; andreas.meryk@i-med.ac.at (A.M.); Werner_Kapferer@hotmail.com (W.K.); beatrix.grubeck@uibk.ac.at (B.G.-L.); 2Department of Pediatrics I, Medical University of Innsbruck, 6020 Innsbruck, Austria; 3Division of Hygiene and Medical Microbiology, Medical University of Innsbruck, 6020 Innsbruck, Austria; marco.grasse@i-med.ac.at; 4Center for Mind/Brain Sciences (CIMeC), University of Trento, 38068 Rovereto, Italy; luigi.balasco@unitn.it

**Keywords:** antioxidants, vitamin C, NAC, immunosenescence, T cells, vaccination, aging

## Abstract

Aging is characterized by reduced immune responses, a process known as immunosenescence. Shortly after their generation, antigen-experienced adaptive immune cells, such as CD8^+^ and CD4^+^ T cells, migrate into the bone marrow (BM), in which they can be maintained for long periods of time within survival niches. Interestingly, we recently observed how oxidative stress may negatively support the maintenance of immunological memory in the BM in old age. To assess whether the generation and maintenance of immunological memory could be improved by scavenging oxygen radicals, we vaccinated 18-months (old) and 3-weeks (young) mice with alum-OVA, in the presence/absence of antioxidants vitamin C (Vc) and/or N-acetylcysteine (NAC). To monitor the phenotype of the immune cell population, blood was withdrawn at several time-points, and BM and spleen were harvested 91 days after the first alum-OVA dose. Only in old mice, memory T cell commitment was boosted with some antioxidant treatments. In addition, oxidative stress and the expression of pro-inflammatory molecules decreased in old mice. Finally, changes in the phenotype of dendritic cells, important regulators of T cell activation, were additionally observed. Taken together, our data show that the generation and maintenance of memory T cells in old age may be improved by targeting oxidative stress.

## 1. Introduction

One of the most dramatic changes in the aging immune system is the involution of the thymus, which leads to a consistent decline in the generation of new naïve T cells [1,2]. As the amounts of naïve T cells are reduced in old age, adaptive immunity is mainly supported by antigen-experienced cells, the maintenance of which is fundamental to fight infections [3]. One typical hallmark of immunosenescence is the accumulation of highly differentiated T cells, known to be detrimental for elderly persons as they are associated with oxidative stress, inflammation, and senescence, which overall supports an increased risk of age-related diseases and mortality [4,5].

Several studies described that the bone marrow (BM) plays an important role in the long-term survival of effector/memory CD8^+^ and CD4^+^ T cells, as well as long-lived plasma cells [6,7]. After clearance of antigens, some newly generated adaptive cells migrate to the BM, where they can be maintained for long periods of time within survival niches [6,8]. In particular, while all antigen-experienced T cells require the cytokine IL-15 for their survival, both memory CD8^+^ and CD4^+^ T cells are maintained by IL-7 [9,10]. Specifically, the survival of terminally differentiated, senescent-like T cells is almost totally based on IL-15, as this subset show high levels of IL-2/IL-15Rβ and low expression of IL-7Rα. In addition, pro-inflammatory cytokine IL-6 is required for the maintenance of highly differentiated T cells.

Recently, we documented that the expression of IL-15 increases, while IL-7 decreases in the human BM in old age, which overall suggests that the survival of bona fide memory T cells may be impaired in the elderly [11,12]. Interestingly, oxidative stress and age-related inflammation (“inflammaging”) were shown to be involved in these changes. In addition, highly differentiated CD8^+^ T cells increased in the aged BM and supported the over-production of oxygen radicals and pro-inflammatory molecules, leading to impaired maintenance of immunological memory as a result [11,13].

In the current study, we investigated whether the phenotype of adaptive immune cells and dendritic cells (DCs), an important regulator of T cell activation, may change in an environment with reduced oxidative stress, using antioxidants N-acetylcysteine (NAC) and vitamin C (Vc). Furthermore, we assessed whether inflammatory parameters in the BM and in the spleen, organ regulating the maintenance and the generation of immunological memory, respectively, may change after administering antioxidants. To achieve this aim, we took advantage of a vaccination protocol using the alum-ovalbumin (OVA) vaccination model, optimal for studying both CD4^+^ and CD8^+^ T cell responses [14]. Young and old mice were vaccinated with three doses of alum-OVA and treated with NAC and Vc, alone or in combination. Only in old mice and in certain conditions, effector/senescent-like CD8^+^ T cells decreased, while memory and memory precursors CD8^+^ T cells increased with antioxidant treatments, in all peripheral blood (PB), BM, and spleen. In addition, oxidative stress in both BM and spleen, and the expression of pro-inflammatory cytokine IL-6 in the spleen decreased in old mice after the treatments. Finally, changes in the phenotype of DCs in the spleen were additionally observed after antioxidant administration.

Taken together, our results show for the first time that the generation and maintenance of memory T cells in old age may be improved after the administration of antioxidants. This may help in planning novel strategies to counteract immunosenescence in the elderly.

## 2. Materials and Methods

### 2.1. Mice

Three-week- and 18-month-old female mice on a C57BL/6J genetic background were maintained under specific pathogen-free conditions at the Institute for Biomedical Aging Research, University of Innsbruck, Austria.

Information about vaccination protocol, antioxidant treatment, bleedings, and organ harvesting are reported in Figure 1. Mice were intraperitoneally (i.p.) injected with Endofit OVA (Invivogen, San Diego, CA, USA; 10 µg/mouse) dissolved in a 1:1 solution of alhydrogel adjuvant 2% (Invivogen) and PBS. Three doses of vaccine were administered to the mice (days 0, 14, and 77). From day −7 (7 days before the administration of the first alum-OVA dose) until day 21, NAC was administered in the drinking water at a concentration of 1 g/L (as reported by Lehmann and coworkers [15]). Vc was i.p. injected, once per day, from day −7 until day 21. Optimal Vc concentration to use in the treatments was chosen after a preliminary experiment (Appendix A). In this titration experiment, the vaccination protocol was performed using a small group of old mice, and the lowest concentration significantly reducing ROS levels in the spleen (10 mg/kg) was chosen.

PB was withdrawn from each mouse on days 7, 14, 21, 28, 42, and 84, and BM and spleen were harvested on day 91. The following experimental groups were included in the experiments, each for 3-week-old and 18-month-old mice: PBS i.p. (control group), NAC, Vc, and NAC+Vc. To ensure reproducibility, the vaccination protocol was repeated twice, in both young and old mice. In both independent experiments, three mice were included in each group, and results from both experiments were pooled, and statistical analysis was performed. One mouse from the groups PBS old (included in experiment 2) and NAC old (experiment 1), and two mice from the group Vc old (one from experiment 1 and one from experiment 2) died before the end of the experiment. No signs of lymphoma or other age-related malignancies were found in the mice reaching the end of the vaccination protocol.

### 2.2. Material Processing and Cell Culture

200 µL blood was harvested from the facial vein and collected in a heparinized tube. Erythrocytes were lysed, incubating the blood with 10 mL lysis buffer (155 mM NH_4_CL, 10 mM KHCO_3_, 0.1 mM EDTA pH 7.4; all Merck KGaA) for 5 min in at RT. After the lysis, blood cells were washed once with RPMI 1640 (Sigma-Aldrich, St. Louis, MO, USA) and resuspended in complete medium (RPMI 1640 supplemented with 10% FCS, 100 U/mL penicillin, and 100 μg/mL streptomycin; Sigma-Aldrich and Invitrogen, Carlsbad, CA, USA, respectively). BM cells were obtained from mice by flushing the femur and tibia with PBS. Spleen cells were digested with Liberase™ Research Grade (Roche, Basel, Switzerland) and DNAse I (Roche) for 30 min, smashed through a Falcon 70 µm cell strainer (Corning, Corning, NY, USA), and washed with complete medium. Isolated cells from the spleen underwent erythrocyte lysis by incubating them with lysis buffer. After the isolation, both BM and spleen cells were washed once with RPMI and resuspended in complete medium.

### 2.3. Flow Cytometry

Immunofluorescence surface staining was performed by adding a panel of directly conjugated antibodies to freshly prepared blood, BM, and spleen cells. To analyze the expression of IL-6, cells were incubated with 10 mg/mL brefeldin A (BFA) for 15 h at 37 °C. To assess the expression of IFNγ, TNF, and IL-2, cells were incubated with 30 ng/mL PMA and 500 ng/mL ionomycin in the presence of 10 mg/mL BFA for 4 h at 37 °C. After surface staining, cells were permeabilized using the Cytofix/Cytoperm kit (BD Biosciences, San Jose, CA, USA), and incubated with the intracellular antibody. Dead cells were excluded from the analysis using 7-AAD or fixable viability dye (FVD) BV421 (both BD Biosciences). Labeled cells were measured using a Fluorescence activated cell sorter (FACS) Canto II (BD Biosciences). Data were analyzed using Flowjo software. The antibodies used in the experiments are shown in Table 1.

### 2.4. ROS Measurement

ROS levels were measured after incubation of BM and spleen cells with the fluorescent dye dihydroethidium (Sigma-Aldrich) at a concentration of 1:250 in complete medium for 20 min at 37 °C.

### 2.5. Statistical Analysis

Statistical significance was assessed after two-way ANOVA and Tuckey post-hoc test, as indicated in the figure legends. A *p* value less than 0.05 was considered significant.

### 2.6. Study Approval

Animal protocols were approved by the Federal Ministry of Science, Research, and Economy of Austria, and carried out in accordance with the Austrian law for animal protection and the institutional guidelines at the University of Innsbruck.

## 3. Results

### 3.1. Effector/Memory T Cell Subsets Change in the Peripheral Blood (PB) after Treatment with Antioxidants

Using the markers IL-7Rα and KLRG-1, the populations IL-7Rα^−^KLRG-1^+^ short-living effector cells (SLEC), and IL-7Rα^+^KLRG-1^−^ memory progenitor effector cells (MPEC) can be identified within CD8^+^ T cells [16,17]. While the first subset may either die soon after the resolution of infections or alternatively accumulate as senescent-like cells, MPEC differentiate into long-living memory CD8^+^ T cells. We investigated the levels of SLEC in the peripheral blood (PB) of untreated and NAC, vitamin C (Vc), and NAC+Vc treated old mice 7, 14, 21, 28, 42, and 84 days after the first alum-OVA injection (Figure 2A). A significant reduction in the frequency of SLEC was observed in Vc treated mice already on day 7, although trends were present also for the other treatments. These differences were stable over the following time-points, and they were significant for all NAC, Vc, and NAC+Vc treatments on day 84. Despite this, no differences between the different treatments were observed. Complete gating strategy is reported in Figure 2. Representative FACS plots are shown in Figure 2C. No differences could be observed between treated and untreated young mice in the levels of SLEC in the PB (Appendix A).

We next assessed whether the treatments may additionally change the frequency of MPEC in the PB of old mice (Figure 2B). Only when Vc alone was administered to the mice, a significant increase of MPEC levels was found on days 42 and 84. Again, no differences were identified in young mice (Appendix A). In addition, the frequency of CD44^hi^CD62L^−^ effector memory (EM) CD8^+^ T cells decreased in the PB of old mice (Figure 2D). Specifically, while NAC treatment reduced EM CD8^+^ T cell levels only on day 84, the treatments with Vc and NAC+Vc were effective in inducing a significant downregulation in the levels of this subset already from day 14. On day 84, NAC and Vc administered alone, but not in combination, induced a consistent reduction in the frequency of EM CD8^+^ T cells. No differences between the treated and untreated groups were observed for EM cells within CD4^+^ T cells in old mice and for EM CD8^+^ T cells in young mice (Appendix A). In addition, the frequency of CD44^hi^CD62L^+^ central memory (CM) CD8^+^ T cells increased in the PB of old mice treated with Vc on day 28, and in mice treated with NAC+Vc on day 84, in comparison with the untreated control (Figure 2D). Increased levels of CM CD4^+^ T cells were additionally found on day 28 in the PB of old mice treated with NAC (Appendix A). No differences were found in young mice, neither for CM CD8^+^ T cells (Appendix A) nor for CM CD4^+^ T cells (data not shown).

Taken together, these results indicate that antioxidant treatments affect the levels of effector/memory T cell subsets in the PB of old mice, particularly within CD8^+^ T cells.

### 3.2. Effector/Memory T Cell Subsets Change in the BM and in the Spleen after Treatment with Antioxidants

We next assessed whether the levels of effector/memory T cell subsets may additionally change in the BM and in the spleen after treatments with NAC, Vc, or NAC+Vc, in comparison with untreated control mice (Figure 3). Similar to the situation in the periphery, the frequency of SLEC in the BM decreased in old mice treated with Vc and NAC+Vc, in comparison with the controls (Figure 3A). While no differences between young and old mice were present in the control groups, the levels of SLEC were significantly lower in the old Vc group, in comparison to young mice treated with Vc. In young mice, no differences were observed after any of the treatments. A similar situation was found in the spleen (Figure 3B). In this case, the levels of SLEC were always higher in old compared to young mice. Again, only in old mice, treatment with Vc and NAC+Vc like in the BM, but also with NAC alone, could significantly reduce the frequency of SLEC. In addition, MPEC increased in old mice treated with NAC and NAC+Vc (both BM and spleen) or Vc alone (only spleen), in comparison with the untreated controls (Figure 3C,D). Interestingly, levels of MPEC in untreated old mice were lower than their younger counterparts. After any of the three treatments, the frequency of this subset was similar between young and old mice. These results indicate that antioxidants may boost MPEC commitment in old mice.

When the frequency of EM CD8^+^ T cells was assessed in the BM, no significant differences were found between all the groups, neither in young nor in old mice (Figure 3E). As reported for SLEC, EM CD8^+^ T cells in the spleen were higher in old compared to young mice and were reduced after NAC, Vc, and NAC+Vc treatments. While no differences were observed for EM CD4^+^ T cells in the BM, this subset was more present in the spleen of old mice, in comparison with any groups of young mice (Appendix A). Despite this, no differences between the treatments were identified. In both BM and spleen, the frequency of CM CD8^+^ T cells was similar in untreated young and old mice (Figure 3G,H). Only in old mice, treatments with NAC or Vc alone (BM), and NAC alone or NAC+Vc (spleen) could increase the levels of this subset, in relationship to untreated controls. Again, no differences were found in young mice. In addition, similar frequencies of CM CD4^+^ T cells were found in all the groups, in both BM and spleen (Appendix A). The expression of activation/exhaustion marker PD-1 was quite low within BM CM CD8^+^ T cells (Figure 3I). Despite this, higher PD-1 levels were found in old compared to young untreated mice, in both BM and spleen (Figure 3I,J). After treatments with NAC and Vc alone, the frequency of PD-1^+^ CM CD8^+^ T cells in old mice was significantly reduced. This was particularly evident in the spleen of old mice treated with Vc, which showed similar PD-1 levels when compared with young-Vc treated mice. No differences were found for co-inhibitory molecules CTLA-4 and TIM-3 (data not shown).

In summary, our data indicate that antioxidants may change the frequency of effector/memory T cell subsets in the BM and in the spleen of old mice.

### 3.3. Antioxidant Treatments Affect ROS Levels and Pro-Inflammatory Molecules in the BM and Spleen

We next investigated whether antioxidant treatments may change the levels of oxygen radicals and pro-inflammatory molecule IL-6 in the BM and in the spleen. As expected, in the untreated groups, ROS levels were higher in old in comparison with young mice, in both BM and spleen (Figure 4A,B). In old mice, after the treatment with NAC alone and NAC+Vc (BM + spleen) and additionally with Vc alone (BM only), oxygen radicals were reduced in comparison with the untreated control group. After any treatments, in both BM and spleen, ROS levels were similar between old and young mice, although no differences were found between treated and untreated young mice. In addition, the expression of IL-6 in the BM was similar between young and old mice, and it did not change after the treatments (Figure 4C, Appendix A). In all experimental groups, IL-6 levels in the spleen were higher in old compared to young mice (Figure 4D, Appendix A). Despite this, treatment with NAC could reduce IL-6 expression in old mice, in relationship to the untreated group.

We then measured the production of IFNγ and TNF, two additional pro-inflammatory and T cell effector molecules, in the BM and in the spleen (Figure 4E–J, Appendix A). In the BM of untreated old mice, more CD8^+^ T cells produced IFNγ in comparison with young mice (Figure 4E). After treatments with NAC, Vc and NAC+Vc, no differences could be found between young and old mice. No age-related differences and no significant differences between the treatments were observed for TNF and IL-2 production in CD8^+^ T cells within the BM (Figure 4F,G). In parallel, more IFNγ, TNF, and IL-2 was produced by BM CD4^+^ T cells from old compared with young mice (Appendix A). Although the treatments did not significantly change the production of IFNγ and TNF, more IL-2^+^CD4^+^ T cells were present in old mice injected with Vc, in comparison with old control mice (Appendix A).

Similar to the situation in the BM, the frequency of IFNγ^+^CD8^+^ T cells in the spleen was higher in old mice, particularly in the groups treated with NAC and Vc alone (Figure 4H). Despite this, no changes were induced by the treatments. When we assessed the production of IFNγ in CD4^+^ T cells from the spleen, higher levels were present in the old group (Appendix A). After Vc treatment, IFNγ production was similar between old and young mice. No differences were observed for the expression of TNF within CD8^+^ T cells (Figure 4I) and CD4^+^ T cells (Appendix A). Reduced IL-2 production was found in the spleen of untreated old mice, in comparison to the young control group. After any of the three treatments, no differences were present between old and young mice. No differences in the expression of IL-2 were present in CD4^+^ T cells from the spleen (Appendix A). Representative FACS plots of IFNγ, TNF, and IL-2 expression within CD8^+^ T cells from BM and spleen are shown in Appendix A.

In summary, our results suggest that, in some conditions, but only in old mice, treatment with NAC and/or Vc could reduce the levels of ROS and affect the production of several T cell cytokines, in the BM and in the spleen.

### 3.4. Dendritic Cell Subsets Change in the Spleen after Antioxidant Treatments

Dendritic cells (DC) are antigen-presenting cells fundamental for the activation of adaptive immune responses. Thus, we investigated whether OVA vaccination in the presence of antioxidants NAC and/or Vc could affect the frequency of certain DC subsets and the phenotype of these cells (Figure 5). The gating strategy reporting the subsets considered in this part of the study is shown in Figure 5A. Classical DC (cDC) are known to express both MHC class II (MHC II) and CD11c, and can additionally be divided into the CD11b^−^ (cDC1) and CD11b^+^ (cDC2) subsets [18].

When we measured all CD11c^+^MHC II^+^ cDCs in the spleen, the frequency was lower in old untreated mice in comparison to the young control group (Figure 5B). NAC+Vc lead to a significant increase in cDC frequency in old mice, and thus, no age-related differences were present after the treatment. In addition, cDC1 cells were higher in young mice in all experimental group, but no significant differences were found after administering the antioxidants (Figure 5C). Furthermore, all NAC, VC, and NAC+Vc treatments lead to an increased frequency of the cDC2 subset, in comparison with old and young untreated groups. Again, no effects were observed when young mice were treated.

Costimulatory molecule CD80 is known to bind its ligands CD28 and CTLA-4 expressed by T cells and regulate the activation of these cells [19]. In cDC1 cells, similar CD80 levels were found in young and old mice, and no differences were present after the treatments (Figure 5E and Appendix A). Only in cDC2 the administration of NAC and Vc, alone or in combination, increased CD80 expression in old mice, in comparison to the untreated group (Figure 5F and Appendix A). Furthermore, the levels of CD40, a costimulatory protein necessary for DC activation [20], were higher in old mice in both cDC1 and cDC2 subsets, particularly in the groups treated with NAC and Vc alone for cDC1 and in all groups for cDC2 (Figure 5G,H and Appendix A). Despite this, no significant changes were induced by the treatments.

Overall, antioxidant treatments may increase the frequency and the activation of cDC subsets in old mice, which may therefore be more efficient in antigen presentation and in activating T cells.

## 4. Discussion

Aging negatively affects the quality of the immune system, which leads to increased frequency and severity of infectious diseases and to a reduced efficiency of vaccinations [21]. Although both innate and adaptive immunity are impaired in old age, T lymphocytes are most severely affected, as they lose the organ involved in their maturation, the thymus [1,2]. Although reduced amounts of antigen-inexperienced, naïve T cells can be observed in the elderly, homeostatic proliferation mechanisms guarantee the presence of a certain amount of naïve T cells, necessary to fight against new infections. Despite this, in old age, naïve T cells may preferentially be committed to an effector, senescent-like T cells, with reduced capability to differentiate into memory cells [22]. In addition, chronic infections, such as cytomegalovirus (CMV), support the accumulation of senescent-like T cells [23]. These cells may accumulate in the BM, therefore reducing the space otherwise available for “true” memory T cells [13,24]. For this reason, finding strategies is helpful—not only to improve the generation, but also the maintenance of memory cells in the BM is fundamental to counteract immunosenescence, and boost immunity against infections in old age.

We recently observed how oxidative stress may impair the long-term maintenance of memory T cells and long-lived plasma cells in the BM [11,13]. In particular, we showed that accumulation of ROS in the BM leads to increased expression of IFNγ and TNF, which induce the production of IL-15 and IL-6, a survival factor for senescent-like T cells. In addition, a negative relationship between Diphtheria antibody concentration in the PB and ROS levels in the BM was identified [24]. Thus, oxidative stress may represent an important feature to target, to test novel strategies aiming at boosting immunological memory.

In our approach, we took advantage of NAC and Vc, well-known antioxidants, already available on the market, and with rare side effects. NAC is known to boost endogenous glutathione levels, as it is a source of amino acid cysteine, which is the rate-limiting step of glutathione synthesis [25,26]. In addition, NAC has shown glutathione-independent antioxidant properties [27]. Vc and glutathione can support each other in the neutralization of ROS. Indeed, reduced Vc (dehidroascorbate, DHA) can rapidly neutralize oxidants, and after this step, it is converted into its oxidized form (ascorbate). Reduced glutathione (GSH) can convert ascorbate into DHA, and thus, after this step Vc is able to react again against ROS. At moderate/high doses, less than 50% of orally administered Vc can reach the bloodstream, while the rest is excreted in the urine [28]. For this reason, we decided to administer this antioxidant i.p. to the mice. To counteract this problem, liposomal Vc formulation has recently been developed, which shows increased absorption rates in comparison to water-soluble Vc [29]. This situation does not happen for NAC, which is very well absorbed when administered orally. Although NAC and Vc were shown to improve immune function in several experimental settings [30,31,32], no studies reported how these molecules affect the phenotype of subsets of immune cells during a vaccination protocol.

We, therefore, established a vaccination schedule in which mice were pre-treated with antioxidants for 7 days, before administering the first alum-OVA dose. In this way, ROS levels may already be lower at the beginning of the vaccination. The treatments with NAC and Vc were stopped on day 21, as mice received two doses of vaccine, and therefore, the generation of memory T cells already started. Finally, a third alum-OVA dose was administered on day 77. This choice has two specific aims: (1) Studying the generation of memory T cells and the phenotype of DC in the spleen two weeks after; (2) investigating the phenotype of T cells migrated into the BM during the first part of the experiment; as memory T cells require 3–8 weeks for the migration to the BM [33], adaptive immune cells generated after the third dose of vaccine can preferentially be found in the spleen. In this way, both the generation and maintenance of immunological memory can be studied with one protocol.

We first assessed the levels of effector/memory T cell subsets in the PB after the injection of alum-OVA. In particular, we investigated whether antioxidants may affect the commitment of naïve T cells into SLEC/EM CD8^+^ T cells, which include short-living cells and/or senescent-like T cells, and MPEC/CM, subsets of memory T cell precursors/memory T cells [16,34].

While the expression of CCR7 and CD62L on CM cells facilitate the homing to secondary lymphoid organs, EM cells preferentially migrate to tissues, are more cytolytic, and express receptors necessary for localization to areas of inflammation [35]. Thus, while SLEC and EM cells show rapid effector functions, and thus, are fundamental when pathogens are present in tissues, MPEC and CM differentiate into long-living memory T cells, and therefore, represent the primary aim of a vaccination in which immunological memory must be triggered. Interestingly, a reduction in the frequency of SLEC/EM and an increased in MPEC/CM CD8^+^ T cells was observed in the PB of old mice treated with antioxidants. While the reduction of SLEC and EM was evident from day 7, the upregulation of MPEC/CM was present only at later time-points. On day 28, an increased frequency of CM CD8^+^ T cells was present in every experimental group. This may be due to administering the second alum-OVA dose, which triggers the generation of new CM cells. In addition, an overall increase of MPEC cells was observed on day 42. CM CD4^+^ T cells additionally increased after NAC treatment, but no differences were observed for EM CD4^+^ T cells. The treatments did not lead to any differences in young mice, which overall showed reduced levels of SLEC and EM CD8^+^ T cells in comparison to untreated old mice. After administering NAC and Vc, similar amounts of circulating SLEC and EM CD8^+^ T cells were found in old and young mice, indicating that the treatment rescued the differences between the two groups. In the spleen of old mice, a consistent decline in both subsets could be observed.

In parallel, MPEC and CM CD8^+^ T cells showed a strong increase. Thus, while without the treatments, old mice produced many more effector cells and less memory CD8^+^ T cell precursors in comparison to young mice, a partial rescue was observed in the treated groups. We can, therefore, speculate that oxidative stress may affect CD8^+^ T cell commitment to memory in old mice. Similar trends were obtained in the BM, in which the differences were less significant in comparison to the spleen. This may be attributable to the fact that effector/memory T cells present in the BM were generated after two doses of vaccine and not after three doses like the ones present in the spleen, and therefore, differences may be attenuated. In addition, the expression of inhibitory receptor PD-1 [36] on CM CD8^+^ T cells, higher in old untreated mice in comparison to young mice, was reduced at the level of young animals after the treatments, in both BM and spleen. This indicates that, in old mice, CM CD8^+^ T cells may be more “ready to react” after NAC and Vc administration.

A key question is whether antioxidant treatment may affect oxidative stress and pro-inflammatory parameters in the BM and in the spleen. In both organs, ROS levels were higher in old untreated mice, which for the BM is in accordance with our data in humans [11]. Interestingly, all antioxidant treatments (BM) and Vc administration (spleen) could rescue this condition, as no differences were present between old and young mice after the treatments. According to the mechanism of action of NAC and Vc, the effects of NAC and Vc together may be additive. Despite this, in our settings, we saw that administering a single antioxidant (either NAC or Vc) may be enough to significantly reduce ROS. Furthermore, we can speculate that the effects of antioxidants may be more stable in the BM in comparison to the spleen, as in this organ, they are still present 70 days after interrupting the treatments. Despite this, we believe that NAC and Vc may act on the spleen environment from the very beginning of their administration, and thus, in the PB, differences can be observed already from day 7. In the spleen, decreased ROS levels were paralleled by a reduction of IL-6 expression. As this cytokine is not only a T cell survival molecule, but also a pro-inflammatory cytokine, we expect that inflammatory processes may be reduced in the spleen of treated mice. No relationship between ROS and IL-6 levels was found in the BM. This is different from the situation in humans and in a small group of SOD1^−/−^ and WT mice, in which a strong correlation between ROS levels and IL-6 MFI in the BM was observed [11].

Furthermore, in the BM, the differences in the expression of IFNγ in CD8^+^ and CD4^+^ T cells between young and old mice disappeared after some of the antioxidant treatments. Thus, we can speculate that inflammation within the BM environment, known to be detrimental for the maintenance of immunological memory in old age [11,13], may partially be attenuated by NAC and/or Vc. Although not completely “rescued” age-related inflammation, commonly known as “inflammaging” and known to support the onset and the severity of age-related diseases, among which immunosenescence [37], may be reduced after antioxidant treatment.

As the last step, we assessed whether the phenotype of DC subsets may be affected by ROS. Overall, in NAC and Vc treated old mice, MHC II^+^ CD11c^+^ cDCs, fundamental for delivering antigens to T cells and regulating the activation of these cells, increased in old mice after antioxidant treatment. In addition, in these mice, administering NAC and Vc lead to increased frequency of cDC2, a subset of migratory DC known to induce potent T follicular helper (Tfh) responses [38]. After the treatment, cDC2 in old mice expressed higher levels of CD80, a costimulatory molecule necessary for the complete activation of T cells. Thus, these results indicate that antioxidants may improve cDC2 functions, which may therefore be more efficient in activating T cells in lymph nodes. Furthermore, activation molecule CD40 was higher in old mice in both cDC1 and cDC2 subsets. Thus, it is important to underline that in old mice, after NAC and Vc administration, the expression of both CD80 and CD40 are higher in comparison to young mice. Overall, this suggests that the capability of priming T cells by DCs may be boosted, reducing oxidative stress in the elderly.

## 5. Conclusions

In summary, our work showed for the first time that commitment of naïve T cells to long-living memory T cells may be boosted by targeting oxidative stress. In addition, the maintenance of immunological memory in the BM may be supported. To achieve this aim, administering one antioxidant (either NAC or Vc) is generally sufficient. Indeed, a certain amount of ROS is required for immune functions [39]. In addition, it is important to underline that, in some conditions, an excessive administration of antioxidants may be even detrimental [40]. Furthermore, our results suggest that the effects of antioxidants are very stable in old mice, as 28 days of treatment led to sustained shifts in T cell subsets and in the production of pro-inflammatory molecules two months later. The administration of NAC and Vc did not lead to any changes in young mice. As these animals showed lower ROS levels in comparison to old mice, antioxidant treatments may be totally ineffective at this age. An optimal age should be identified, to start the treatments to target ROS and boost the functionality of the immune system. Thus, ROS scavengers administered at the right timepoint may represent an optimal tool to support not only the generation of new memory T cells, but also the maintenance of adaptive immune cells generated years before and maintained in the BM niches.

As NAC and Vc seem to boost immune responses in old age, future studies must assess the impact of antioxidants on a vaccination protocol performed using an antigen from a pathogen relevant for aging, such as a mouse-adapted influenza virus strain. Indeed, it is known that the efficacy of influenza vaccination is reduced in the elderly [41], and therefore, it will be important to understand whether antioxidants may be effective also in this context. The results obtained in our study using alum-OVA will certainly help to define optimal settings for future vaccination protocols.

In addition, the effects of NAC and Vc on levels of pro-inflammatory cytokines in the plasma will be analyzed in future studies. Furthermore, it must be assessed whether the long-term exposure to antioxidants may change molecular pathways controlling T cell differentiation, such as JAK/STAT, MAPK signaling, and HIF1α stabilization.

Finally, nutritional intervention, including natural antioxidants and/or nutraceutical compounds, must be tested to assess whether a healthy lifestyle may additionally boost the generation and maintenance of immunological memory in the elderly.

## Figures and Tables

**Figure 1 antioxidants-09-01152-f001:**
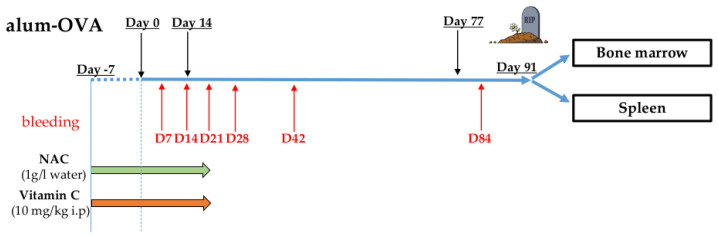
Vaccination protocol (see Materials and Methods). Mice were intraperitoneally (i.p.) injected with three doses of alum-OVA on days 0, 14, and 77. From day −7 until day 21, NAC was administered in the drinking water, and Vc and PBS (control group) were i.p. injected once per day. Blood was withdrawn from each mouse on days 7, 14, 21, 28, 42, and 84, and BM and spleen were harvested on day 91.

**Figure 2 antioxidants-09-01152-f002:**
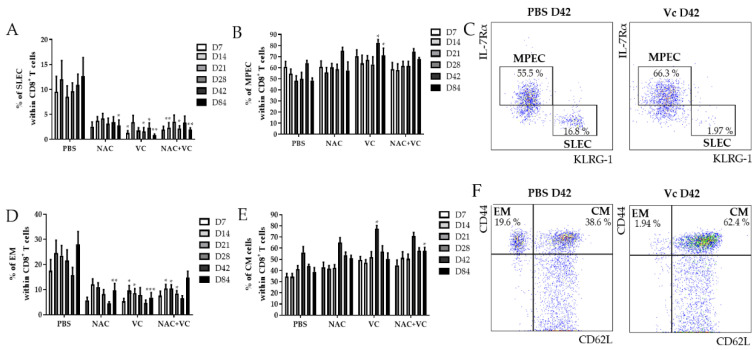
Effector/memory CD8^+^ T cell subsets in the PB of old mice. Frequency of (**A**) IL-7Rα^−^KLRG-1^+^ SLEC and (**B**) IL-7Rα^+^KLRG-1^−^ MPEC in the blood of old mice treated with PBS, NAC, Vc, or NAC+Vc. (**C**) Representative FACS plots of SLEC and MPEC in a control and in a Vc-treated old mouse on day 42. Frequency of (**D**) CD44^hi^CD62L^−^ EM and (**E**) CD44^hi^CD62L^+^ CM CD8^+^ T cells in the blood of old mice treated with PBS, NAC, Vc, or NAC+Vc. (**F**) Representative FACS plots of SLEC and MPEC in a control and in a Vc-treated old mouse on day 42. Blood was harvested on days 7, 14, 21, 28, 42, and 84. Data are shown as mean ±SEM. Two-way ANOVA, Tukey post-hoc test. * *p* < 0.05; ** *p* < 0.01; *** *p* < 0.001.

**Figure 3 antioxidants-09-01152-f003:**
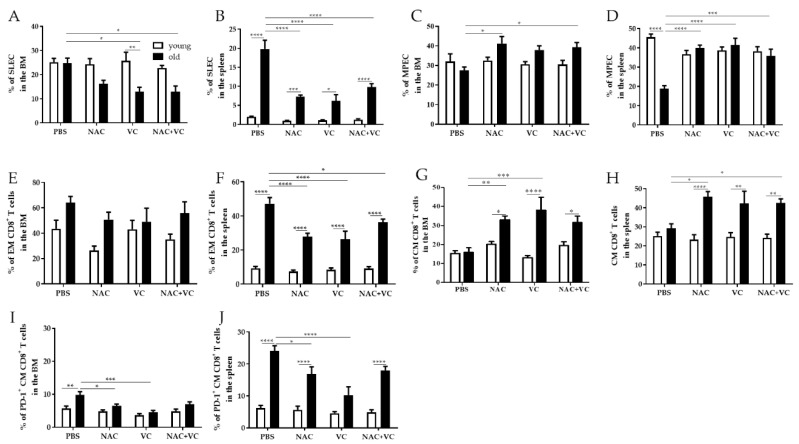
Effector/memory CD8^+^ T cell subsets in the BM and in the spleen. Frequency of (**A**) SLEC in the BM, (**B**) SLEC in the spleen, (**C**) MPEC in the BM, (**D**) MPEC in the spleen, (**E**) EM CD8^+^ T cells in the BM, (**F**) EM CD8^+^ T cells in the spleen, (**G**) CM CD8^+^ T cells in the BM, (**H**) CM CD8^+^ T cells in the spleen, (**I**) PD-1^+^CM CD8^+^ T cells in the BM, (**J**) PD-1^+^CM CD8^+^ T cells in the spleen of young (white columns) and old (black columns) mice treated with PBS, NAC, Vc, or NAC+Vc. Data are shown as mean ± SEM. Two-way ANOVA, Tukey post-hoc test. * *p* < 0.05; ** *p* < 0.01; *** *p* < 0.001; **** *p* < 0.0001.

**Figure 4 antioxidants-09-01152-f004:**
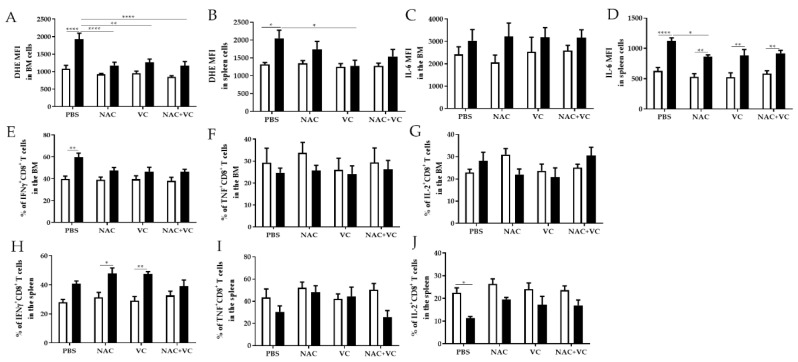
ROS levels and pro-inflammatory molecules in the BM and spleen. (**A**) ROS levels (=DHE Mean Fluorescence Intensity, MFI) in the BM, (**B**) ROS levels in the spleen, (**C**) IL-6 MFI in the BM, (**D**) IL-6 MFI in the spleen, (**E**) IFNγ^+^CD8^+^ T cells in the BM, (**F**) TNF^+^CD8^+^ T cells in the BM, (**G**) IL-2^+^ CD8^+^ T cells in the BM, (**H**) IFNγ^+^CD8^+^ T cells in the spleen, (**I**) TNF^+^CD8^+^ T cells in the spleen, (**J**) IL-2^+^ CD8^+^ T cells in the BM of young (white columns) and old (black columns) mice treated with PBS, NAC, Vc, or NAC+Vc. Data are shown as mean ± SEM. Two-way ANOVA, Tukey post-hoc test. * *p* < 0.05; ** *p* < 0.01; **** *p* < 0.0001.

**Figure 5 antioxidants-09-01152-f005:**
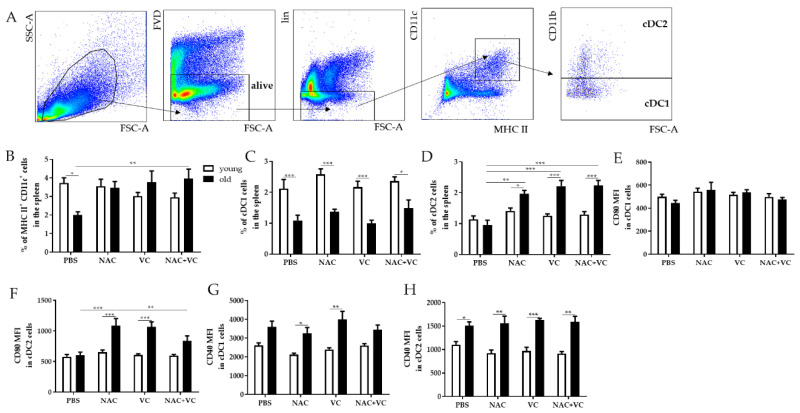
Dendritic cell subsets in the spleen. (**A**) Gating strategy for DC subsets. After excluding dead cells using the FVD Zombie Violet, lineage negative (lin-) CD3^−^NKp46^−^CD19^−^ cells were considered. FACS plots showing the gating for CD11c^+^MHC II^+^ DCs, CD11c^+^MHC II^+^CD11b^−^ classical DCs (cDC1) and CD11c^+^MHC II^+^CD11b^+^ DCs (cDC2) are reported. Frequency of (**B**) CD11c^+^MHC II^+^ DCs, (**C**) cDC1, (**D**) cDC2, and MFI of (**E**) CD80 in cDC1, (**F**) CD80 in cDC2, (**G**) CD40 in cDC1, and (**H**) CD40 in cDC2 in the spleen of young (white columns) and old (black columns) mice treated with PBS, NAC, Vc, or NAC+Vc. Data are shown as mean ± SEM. Two-way ANOVA, Tukey post-hoc test. * *p* < 0.05; ** *p* < 0.01; ****p* < 0.0001.

**Table 1 antioxidants-09-01152-t001:** Antibodies used in the FACS stainings.

Antigen	Fluorocrome	Company	Clone
KLRG-1	BV421	Biolegend	2F1/KLRG1
CD44	FITC	Biolegend	IM7
IL-7Rα	PE	Biolegend	A7R34
CD8	PeCy7	Biolegend	53-6.7
CD8	PerCp	Biolegend	53-6.7
CD62L	APC	Biolegend	MEL-14
CD4	BV510	Biolegend	GK1.5
CD4	Pecy7	Biolegend	GK1.5
CD3	APC-Vio770	Miltenyi	REA641
PD-1	PE	Biolegend	29F.1A12
IL-6	APC	Biolegend	MP5-20F3
IFNg	PE	Biolegend	XMG1.2
TNF	FITC	Biolegend	MP6-XT22
IL-2	APC	Biolegend	JES6-5H4
CD3	BV510	Biolegend	17A2
CD11c	BV510	Biolegend	N418
MHC II	FITC	Biolegend	M5/114.15.2
CD40	PE	Biolegend	3/23
NKp46	PerCp	Biolegend	29A1.4
CD19	PerCp	Biolegend	6D5
CD3	PerCp	Biolegend	145-2C11
CD80	APC	Biolegend	16-10A1
CD11b	APC-Cy7	Miltenyi	M1/70.15.11.5

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
