# Peer review of "Antioxidants N-Acetylcysteine and Vitamin C Improve T Cell Commitment to Memory and Long-Term Maintenance of Immunological Memory in Old Mice"

_antioxidants, 2020, doi:10.3390/antiox9111152_

Round 1

Reviewer 1 Report

The authors have made a great effort to improve the quality of the manuscript. All of the concerns in the previous comments have been properly addressed.

Author Response

We thank the reviewer for the comment.

Reviewer 2 Report

The authors of the manuscript entitled “ Antioxidants N-acetylcysteine and vitamin C 3 improve T cell commitment to memory and long-term maintenance of immunological memory in old mice” have provided replies to the comments raised by the reviewers. The manuscript has improved significantly. Despite the lack of assessment of antigen-specific responses, the findings are interesting and promising enough to ensure publication. However, there are some minor aspects that still need to be addressed before it can be published.

Minor issues

Since I assume the IL-6 data is available as MFI and % of the parent cells. I suggest the authors add the % of parent cells data to supplementary information. In this way, the study will be comparable to previous studies from this group and also display the data in the same format of the other cytokines in the current manuscript.

Author Response

Following the suggestion of the reviewer, the % of parent cells data for IL-6 have been added to the supplementary information (new Supplementary Figure 5).

This manuscript is a resubmission of an earlier submission. The following is a list of the peer review reports and author responses from that submission.

Reviewer 1 Report

In this study, Andreas Meryk et al. studied the effect of N-acetylcysteine (NAC) and vitamin C (Vc) on alum-OVA-treated mice. The overall manuscript is standard organized but written with several typographic and grammatical errors or overlooked errors, which reduces the readability and quality of the article. An English correction must be made. 

Author Response

After the suggestion of the reviewer, corrections have been made and the quality of the manuscript has now been improved.

My questions and comments:

You studied the effect of NAC and Vc on aging/immunosenescence and you used alum-OVA model but I lack information that this model is commonly used for these purposes. Please add this information into introduction part because it is normally used to sensitize specific immunity.

Author Response

In this study we decided to choose a standard antigen (OVA) commonly used to study T cell responses, rather than focusing on a specific antigen more relevant for aging (see reviewer’s 3 comments). Our aim was initially to assess whether immune responses could be enhanced using antioxidants, in both young and old mice. No studies used this model to study T cell responses in the context of immunosenescence. Despite this, the appropriateness of alum-OVA for studying T cell responses is proven (for example by McKee et al., 2009). As suggested by the reviewer, we added more information about this model in the introduction part (lines 60-63). The possibility of using antigens more relevant to aging is now discussed (discussion part, lines 435-440)

- Please check and correct all abbreviation in text (for example PB line 64; 126 and 131)

Author Response

All abbreviations in the text have been checked and corrected.

The paragraph “Mice” in section Materials and methods is little bit misunderstood. Therefore, this part must be rewritten.

Author Response

The paragraph has now been rewritten and made it clearer. 

- one of the misleading fact: line 80-81: “and Vc or PBS (control) was i.p. injected, once per day, at a concentration of 10 mg/kg.” How did you reach the 10 mg/kg PBS concentration?

Author Response

We introduced a new Supplem. Figure (new Suppl. Figure 1) showing data about the Vc titration, performed on a small group of old mice. Briefly, mice were treated from day -7 until day 21 with 4 doses of Vc (1, 10, 100 and 300 mg/kg) and ROS levels in the spleen were measured on day 91. In addition, a reference for the concentration used for NAC has been added.

- please check and correct: line 62: “and, andtreated”; line 64 “memory/memory precursors”; line 78 “10u/mouse”

Author Response

The corrections have been made.

- check and correct all superscripts and subscripts (for example: line 94)!!

Author Response

Superscripts and subscripts have been checked and corrected.

- check and correct text formatting mainly font (for example line 155-156; line 200-201 or line: 278-279)!!

Author Response

Text formatting has been corrected.

-  Figure 2E: how do you explain or what caused in 28 days (the results are higher than others)?

Author Response

We believe that the increased frequency of CM on day 28 in every experimental group may be due to the administration of the second dose of alum-OVA, which triggers the generation of new CM cells. This increase on MPEC is visible only on day 42, and thus this subset may take a bit longer to be released by the spleen. We thank the reviewer for this point, we now added it into the discussion part (lines 365-368)

- Please, delete the “Figure 5” from right-bottom corner of Figure 5 (line 280).

Author Response

“Figure 5” has now been deleted.

- How do you explain situation/mechanism of action of NAC and Vc combination? If the effect were dependent on antioxidant activity, then we should observe same or additive effect.

Author Response

According to the mechanism of action of NAC and Vc (see the answer to the last point), the effects of NAC and Vc together may be additive. Despite this, in the BM (Figure 4A) we saw that the single antioxidants were enough to significantly reduce ROS levels. In the spleen, only Vc administered alone could significantly reduce ROS, (although trends were present also for the other conditions). From these data we can assume that the administration of a single antioxidant (either NAC or Vc) may be enough to significantly reduce ROS. Indeed, in some conditions, an excessive administration of antioxidants is known to be detrimental (as reported by Seifirad et al.). Furthermore we can speculate that the effects of antioxidants may be more stable in the BM, in which it is still present 70 days after interrupting the treatments, in comparison to the spleen. This point has now been added to the Discussion part (lines 390-395)

- Longer-term exposure is likely to have a number of other effects, such as a change in the JAK/STAT or MAPK signaling pathways or HIF-alpha stabilization which could lead to change the differentiation.

Author Response

This aspect has now been added to the discussion part (lines 442-444)

- During OVA sensitization, the spleen enlarges and lymphocytes proliferate and differentiate. Did you measured other parameters of spleen (like weight, ratio of others white bloods, redox enzyme SOD, CAT or other)?

Author Response

Unfortunately we did not measure any of these parameters in the spleen. Despite this, no differences in spleen size have been observed in the mice. Although we did not directly measure SOD and CAT activity, DHE MFI indirectly gives information about SOD activity as the dye reacts with O2•− present in the cells.

- Only IL-6 level was significantly decreased in spleen. Did you detected some pro-inflammatory cytokine in plasma? In fact, used long-exposure of antioxidants play a key role in whole body of mice not only in spleen and bone marrow.

Author Response

We measured IL-6 only in the spleen and in the BM, as we wanted to assess whether antioxidants may affect pro-inflammatory parameters in these organs involved in the generation and maintenance of immunological memory. We strongly agree that assessing the levels of pro-inflammatory cytokines in the plasma (and eventually also in other organs) after antioxidant treatment is of interest and it will certainly be done in future experiments. A sentence about this topic has been added to the discussion part, lines 441-442

- I miss the information on the mechanism of action of these antioxidants.

Author Response

NAC is known to boost endogenous glutathione levels, as it is a source of aminoacid cysteine which is the rate-limiting step of glutathione synthesis (Lasram, Rushworth). In addition, NAC has shown glutathione-independent antioxidant properties (Sun et al). Vitamin C and glutathione are known to act synergistically in the neutralization of ROS. Reduced vitamin C (dehidroascorbate, DHA) can rapidly neutralise oxidants, and after this step it is converted into its oxidized form (ascorbate). Reduced glutathione (DHA) can convert ascorbate into DHA, and thus after this step vitamin C is able to react again against ROS. In this way, an additive effect of NAC and Vc can be expected. We now added this information into the discussion part (lines 328-334)

Reviewer 2 Report

Meryk et al present work on the impact of antioxidants, NAC and Vitamin C alone or combined, on various immune parameters in the bone marrow and the spleen. T cell subsets are assessed for TEM/TCM and MPEC/SLEC frequencies in the spleen, bone marrow and blood, and ROS and cytokine production in the spleen and bone marrow, with additional analyses of DC subsets and activation in the spleen.

My main concern with this data set is reproducibility. It isn’t clear to me how many mice were used for each analysis (in the methods it refers to 6 mice per treatment group, as well as n=3) and how many times this was repeated (it doesn’t appear to have been repeated).

Author Response

We have probably been misunderstood by the reviewer. Our experiment was performed twice (in the case of Vc even 3 times, although we used these data only for the Vc titration as reported in Suppl. Figure 1), and in both cases 3 mice were included in each experimental group. In this way, 6 mice were present in each group, but they were pooled from 2 independent experiments. We now clarified this in the Material and Methods part, paragraph “mice”.

 In addition, the methods note that 2 mice died from one group and 1 mouse from 2 groups during the time course. My concern is due to the fact that individual ageing mice are very variable- they develop co-morbidities (such as lymphoma commonly around the 18-20 month age bracket in C57BL/6 mice) and these co-morbidities can substantially skew differentiation phenotypes such as TEM/TCM and SLEC/MPEC profiles. It therefore becomes important to note, were signs of lymphoma (enlarged spleens) or other malignancy observed in the mice that were harvested at day 91 for BM and spleen tissue, and were these mice excluded? As a result of this inherent variability and working with low numbers of biological replicates, being able to repeat findings in aged mice with a reasonable number of individual animals becomes very important. This work should be repeated at least once more before publication- mice with co-morbidity in the PBS group could conceivably lead to the observed differences and it is important to rule this out.

Author Response

As correctly said by the reviewer, comorbidities play an important role in old age and when present may affect immunological parameters. Despite this, in our settings, no signs of lymphoma or other malignancies were found in the mice which reached the end of the vaccination protocol. In every animal, spleen showed similar sizes. Comorbidities may probably have been present in the mice that died before day 91. As a proof of concept, values for each measured parameter are quite homogeneous within each group (even after pooling the values from two independent experiments), and standard deviations are not excessively high. Thus, having performed the experiment twice to ensure reproducibility, we don’t believe that repeating the experiment a third time would introduce consistent differences in our results. To clarify this aspect, we now introduced the sentence “No signs of lymphoma or other age-related malignancies were found in the mice reaching the end of the vaccination protocol” in the Material and Methods (lines 94-96)

In terms of the blood profile, substantial shifts in TEM/TCM and MPEC/SLEC frequencies appear to have been achieved with as little as 14 days of treatment. However, animals were not bled prior to treatment, so it is difficult to assess whether the PBS treated aged group had a different baseline or antioxidant treatment had that much of an impact relatively quickly.

Author Response

As correctly said by the reviewer, bleeding was performed only after the beginning of the vaccination protocol as we aimed at following the release of memory progenitor/memory/ effector T cells from the spleen. Performing the same analysis on naïve unvaccinated mice would not have been optimal, as in this case T cell responses were not triggered by OVA. Despite this, we can hypothesize that antioxidant may act on the spleen environment, and thus the differences that we see in the blood reflect the situation present in the spleen.  A sentence about this topic has now been added to the discussion part (lines 395-397)

In addition, the antioxidant treatment was stopped at d21, so it appears that 28 days of antioxidant treatment led to sustained shifts in T cell subsets 2 months later- this would be a remarkable outcome but it warrants repeating (see discussion above) to validate.

Author Response

We agree with it and we now introduced a sentence in the discussion part (lines 426-429) to make this aspect more evident to the reader. As already mentioned before, our experiment was validated as it was performed twice

One thing that was confusing is that, despite vaccinating with OVA-alum, this paper does not evaluate OVA-specific T cell responses. The peripheral blood analyses of differentiation states are performed in polyclonal T cells and the analyses in the spleen and bone marrow are performed after polyclonal stimulation. Given that OVA-Alum doesn’t induce a massive CD4 and CD8 T cell response in C57BL/6 mice, it is unlikely that the vaccination is driving the documented changes, which appear to be in total T cells. If the authors have the opportunity to repeat the work (see discussion above), I would suggest that valuable supplemental data would be to restimulate with OVA protein (for CD4 cell responses) and SIINFEKL peptide (for CD8 T cell responses).

Author Response

In our study we assumed that immune responses were induced by OVA antigen, which is a logic assumption as naïve mice housed in animal facilities do not show ongoing immune responses that are as strong as the OVA-specific ones. Such big amount of memory (CD44+) CD8+ and CD4+ T cells would not have been generated without a vaccination protocol with a strong antigen like the one we used. Thus, we decided to use PMA/Iono/BFA stimulation in order to achieve a stronger stimulation of our cells in comparison to OVA peptides.

Reviewer 3 Report

The manuscript entitled “ Antioxidants N-acetylcysteine and vitamin C 3 improve T cell commitment to memory and long-term maintenance of immunological memory in old mice” by Meryk et al provide interesting data regarding the potential use of the antioxidants( NAC and Vitamin C) to improve memory T cells in aged mice. Provocative data is shown regarding the immunophenotype changes by antioxidants in the MPEC/SLEC as well as in the EM/CM subsets in the aged mice. It is also interesting that the researched showed a reduction in ROS in the aged animals that received antioxidants. Despite this, there are multiple aspects that need to be addressed before it can be accepted for publication:

Major issues

OVA is an ok antigen, but the manuscript would be more relevant if an antigen from a pathogen relevant for this age group is included. For example, elderly humans are recommended to be vaccinated for influenza virus, making antigen relevant for this age group. There are several mouse-adapted influenza virus strains that can be easily included in the experimental design of the manuscript and that would give relevance to the data presented. Also, influenza virus allows the use of a challenge system that can provide even more relevant data. Antigens from other pathogens relevant for this age group could be assess instead of influenza virus.

Author Response

As previously discussed (see response to reviewer 1), for this study we decided to choose a standard antigen (OVA) commonly used to study T cell responses, and we did not focus on a specific antigen more relevant for aging. Before performing the experiments described in the current manuscript, we did not expect that antioxidants may lead to such consistent effects specifically in old mice. Indeed, we wanted simply to assess whether antioxidant treatments may show some effects on immune responses in general, using mice with different ages. We believe that this represents the first step before performing a more detailed analysis. As a proof for the validity of the chosen model, our study was positively judged by external reviewers of the field, and we obtained a Tyrolean Science Funds (TWF, grant number ZAP746006) as a financial support for this project.

Therefore, we thank the reviewer for the suggestion and, having now assessed that antioxidants strongly influence immune responses in old age, future studies will certainly focus on pathogens more relevant for aging. Despite this, we believe that our data are a good piece of information, and important for planning more targeted studies. We added a sentence about this in the Discussion part (lines 435-440)

IL-6 data is presented as MFI. It is not clear why the data is presented in this format. It would be more relevant if it is presented as percentage of CD4 and CD8 T cells producing this cytokine, similar to data presented for IFN-g, TNF and IL-2.

Author Response

In this study we measured IL-6 MFI as we did in another recent paper (Pangrazzi et al., EJI, 2017). In this way, a direct comparison between the two studies was possible. As IL-6 is a key pro-inflammatory marker, measuring the expression of this cytokine in the whole BM and spleen environment gives an indication of inflammatory processes within these organs. We now added two sentences to the discussion (lines 399-402)

Dendritic cell gating not appropriate. The cells should be gated at the very least as cDC1 (CD11c+ MHCII+ CD11b-) and cDC2 (CD11c+ MHCII+ CD11b+). Other markers might improve definition of these populations (e.g., XCR1). CD11b alone cannot be used to define a DC subset. Therefore, all the discussion of DCs need to be adjusted after the cells are gated appropriately.

Author Response

Following the suggestions of the reviewer, we changed our gating strategy and we analysed the frequency of cDC1 and cDC2 and the expression of CD80 and CD40 within these subsets. The parts about DCs have been changed accordingly (results part par. 3.4 and discussion part, lines 409-420)

Minor issues

While NAC was given orally, it is not clear why Vitamin C was not provided orally as well. The most natural way to supplement Vitamin C in humans would be Vitamin C; therefore, a rational for IP use is necessary.

Author Response

Although, as correctly said by the reviewer, an oral Vc administration would be the most natural way to supplement this antioxidant in humans, its absorption is low in this form. At moderate/high Vc doses, less than 50% of this molecule can reach the bloodstream while the rest is excreted in the urine (Jacob et al). This situation does not happen for NAC, which is very well absorbed when administered orally. To counteract this problem, liposomal Vc formulation has recently been developed, which shows increased absorption rates in comparison to water-soluble Vc (Lukawski). We added this point to the Discussion part, (lines 334-339)